# Effects of growth promoting microorganisms on tomato seedlings growing in different media conditions

Robert Pokluda[1]*, Lucia Ragasová[1], Miloš Jurica[1], Andrzej Kalisz[2], Monika Komorowska[2], Marcin Niemiec[3], Agnieszka Sekara[2]

1 Faculty of Horticulture, Department of Vegetable Sciences and Floriculture, Mendel University in Brno, Brno, Czech Republic, 2 Faculty of Biotechnology and Horticulture, Department of Horticulture, University of Agriculture in Krakow, Krakow, Poland, 3 Faculty of Agriculture and Economics, Department of Agricultural and Environmental Chemistry, University of Agriculture in Krakow, Krakow, Poland

* pokluda@mendelu.cz

**Data Availability Statement:** All relevant data are within the manuscript.

## Abstract

Plant growth-promoting microbes (PGPM) play vital roles in maintaining crop fitness and soil health in stressed environments. Research have included analysis-based cultivation of soil-microbial-plant relationships to clarify microbiota potential. The goal of the research was to (i) evaluate the symbiotic microorganism effects on tomato seedling fitness under stressed conditions simulating a fragile soil susceptible to degradation; (ii) compare the plant-microbial interactions after inoculation with microbial isolates and fungi-bacteria consortia; (iii) develop an effective crop-microbial network, which improves soil and plant status. The experimental design included non-inoculated treatments with peat and sand at ratios of 50:50, 70:30, 100:0 (v:v), inoculated treatments with arbuscular mycorrhizal fungi (AMF) and *Azospirillum brasilense* (AZ) using the aforementioned peat:sand ratios; and treatment with peat co-inoculated with AMF and *Saccharothrix tamanrassetensis* (S). AMF + AZ increased root fresh weight in peat substrate compared to the control (4.4 to 3.3 g plant$^{-1}$). An increase in shoot fresh weight was detected in the AMF + AZ treatment with a 50:50 peat:sand ratio (10.1 to 8.5 g plant$^{-1}$). AMF + AZ reduced antioxidant activity (DPPH) (18–34%) in leaves, whereas AMF + S had the highest DPPH in leaves and roots (45%). Total leaf phenolic content was higher in control with a decreased proportion of peat. Peroxidase activity was enhanced in AMF + AZ and AMF + S treatments, except for AMF + AZ in peat. Microscopic root assays revealed the ability of AMF to establish strong fungal-tomato symbiosis; the colonization rate was 78–89%. AMF + AZ accelerated K and Mg accumulation in tomato leaves in treatments reflecting soil stress. To date, there has been no relevant information regarding the successful AMF and *Saccharothrix* co-inoculation relationship. This study confirmed that AMF + S could increase the P, S, and Fe status of seedlings under high organic C content conditions. The improved tomato growth and nutrient acquisition demonstrated the potential of PGPM colonization under degraded soil conditions.

**Funding:** This paper was supported by project of Internal grant agency of Faculty of Horticulture, Mendel University in Brno IGA - ZF/2020 - AP004 (RP). Microscopic analyses were done on infrastructure supported by project OP VVV CZ.02.1.01/0.0/0.0/16_017/0002334. Research infrastructure for young scientists, financed from structural funds of EU and Ministry of education of the Czech Republic (RP). The paper was supported by Polish-Czech Joint Research Project PPN/BCZ/ 2019/1/00014; Microorganisms biodiversity impact on soil conservation in agricultural crop system (AS). The funders had no role in study design, data collection and analysis, decision to publish, or preparation of the manuscript.

**Competing interests:** The authors have declared that no competing interests exist.

## Introduction

The predicted and persistent threat of soil degradation driven by climatic and anthropogenic forces necessitates the development of strict directives for the protection of soils, as recently promoted by the European Union [1]. Erosion, and the consequent decline in organic soil matter, reduces overall soil biomass and biological activity. Ultimately, it profoundly affects the diversity of microbes coexisting in the soil ecosystem [2,3]. Understanding the soil microbial communities and their beneficial roles in agricultural land is crucial to maintaining plant fitness and preventing soil degradation [4–8]. The rhizosphere is home to many interactions that substantially affect soil, microorganisms, and plants. In agricultural practices, root-associated arbuscular mycorrhizal fungi AMF application is highly effective, ultimately resulting in the improvement of crop growth, health, yield, and general fitness [9–11]. Nutrient availability, especially phosphorus (P), is essential for multiple processes. P is delivered as phosphate to the root through AMF specialized structures called arbuscules, formed within cortical cells [12]. Volatile compounds released by rhizosphere microorganisms are also reported to be signal molecules stimulating lateral root development [13]. Fungi release carbolic acid, and low-molecular-weight organic acids, that bind to metal ions in the soil solution, including Al, Ca, Cu, Fe, Mg, Mn, and Zn ions, promoting mineral weathering and enabling their absorption by hyphae and transport to plant roots [12]. Enhanced nutrient uptake, mainly that of N and P, is rewarded with C compounds derived from host plant photosynthesis [14]. The second area of AMF action is the increased tolerance of the host plant to biotic and abiotic stressors, more effective nutrient and water absorption, higher photosynthetic activity, control of reactive oxygen species by increased activity of antioxidant enzymes [15–17]. AMF covers some basic (such as the alteration in root morphology, increased plant nutrition, and damage compensation) and secondary phenomena (competition between symbiotic and parasitic microorganisms, changes in rhizosphere microbial populations, and the activation of plant defense mechanisms) [18,19]. The phylum *Actinobacteria* represents a large group of non-mycorrhizal plant growth promoting microorganisms (PGPM), among them *Saccharothrix* spp. are aerobic, gram-positive actinomycetes with branching vegetative mycelium, that fragments into rod-shaped spores [20]. Merrouche et al. [21] have previously reported the intense activity of *Saccharothrix* against fungi (e.g., *Fusarium* spp.) and moderate activity against bacteria. PGPM and their positive impact on the plant have not been as broadly studied as AMF, although some studies have reported their beneficial effects on crops [22]. PGPM can shape the relationships in the rhizosphere microbiome through P solubilization, or induction of plant stress tolerance [23]. The significant beneficial effects of *Azospirillum* includes the capacity to fix atmospheric N, synthesize phytohormones and plant regulators, and increase plant tolerance to abiotic and biotic stresses [24]. Previous efforts to significantly change the indigenous microflora of the soil by introducing single cultures of extrinsic microorganisms have not always been successful. Thus, the probability of shifting the "microbiological equilibrium" of the rhizosphere and controlling it to favor the growth, yield, and health of crops is much greater if consortia of beneficial and effective microorganisms are introduced that are physiologically and ecologically compatible with one another [25–27]. When the consortia of AMF and PGPM become established, their individual beneficial effects are often magnified in a synergistic manner [28–30], although the relationships are not always simple. Clearly, the direct effects of PGPM on the trade-off among microbes and plants are still poorly understood regarding soil and crop fitness.

According to Pedersen et al. [31], agriculturally fit crops are defined as the most useful to humans in agricultural systems and the food industry. Tomato is a highly mycorrhizal species, and PGPM can act as stimulants for the growth and development of tomato plants directly or

indirectly via the availability of many essential nutrients and phytohormones, and the suppression/destruction of plant diseases, decreasing oxidative stress, and activation of pathogenesis-related metabolites, as it had been reported for other crops [32–36]. Inui Kishi et al. [37] stated that PGPM can be successfully applied to manage crop fitness in degradable areas, because PGPM elicit so-called 'induced systemic tolerance' to salt, drought, and nutrient deficiency in rhizosphere, so tomatoes will benefit from PGPM when cultivated under easily degradable soil. The unresolved problem is that the PGPM-host relationships are highly dynamic and can evolve from symbiosis to parasitism when net costs of the symbiosis exceed net benefits [38]. Moreover, the symbionts enhance their own fitness, not necessarily the fitness of their host [32]. Thus, inoculation may result in a biotic stressor that triggers defense mechanisms in the plants. The analysis of plant physiological status and biochemical stress biomarkers can explain the nature of the reaction of the plant to inoculation under specific conditions of easily degradable soil. An innovative approach uses bacteria and fungi isolates and their consortia to directly link the effect to the microorganism species to understand the principles of the more complex biotrophic interactions [39]. The goal of the study was to evaluate the symbiotic microorganism effects on plant development under stressful conditions of soil susceptible to degradation. Specific goals include (i) evaluation of symbiotic microorganism effects on tomato seedling fitness under stressful conditions simulating those of soil susceptible to degradation; (ii) comparison of plant-microbial interaction after inoculation of microbial isolates and fungi-bacteria consortia; and (iii) development of an effective crop-microbial network, for the improvement of soil and plant status.

## Materials and methods

### Material and treatments

The experimental plant was the tomato *Solanum lycopersicum* L. cultivar Spencer $F_1$ (Moravoseed Ltd., CZ). Tomato seedlings were cultivated in a growth chamber in pots filled with the different types of autoclaved (120 ˚C for 60 min) substrates. Three mixtures of sowing peat (Klasmann, DE) and sand (local source) were used, namely, peat:sand ratios of (v:v) (i) 100:0, (ii) 70:30, and (iii) 50:50. The pH was maintained at 6.5 by the use of calcium carbonate in relevant amounts. Selected substrate parameters are shown in Table 1. The substrates were inoculated with beneficial microorganisms: (i) arbuscular mycorrhizal fungi mix (AMF), (ii) AMF fungi and *Azospirillum brasilense* CCM 3862 (AMF + AZ), and (iii) *Saccharothrix tamanrassetensis* SA 198 (AMF + S). AMF mix contained *Claroideoglomus claroideum* BEG 96, *Claroideoglomus etunicatum* BEG 92, *Funneliformis geosporum* BEG 199, *Funneliformis mosseae* BEG 95, and *Rhizophagus irregularis* BEG 140, and was applied as an AMF mixture containing 145 spores per g at a dose of 0.015 g $cm^3$ of the substrate. The AMF + AZ treatment contained AMF and *A. brasilense* ($10^8$ CFU in sterile 1XPBS). Seeds were soaked for 30 min. and were used for tomato inoculation with *A. brasilense*. *S. tamanrassetensis* was grown in a yeast-malt

**Table 1. Selected substrate parameters at the beginning of the experiment.**

| Peat: sand treat-ment | pH | Dry matter | N-NO₃ | N-NH₄ | N total | K | P | Mg | Na | Ca | Cation exchange capacity | Soil weight per pot | Substrate bulk density |
|---|---|---|---|---|---|---|---|---|---|---|---|---|---|
| | CaCl | % | mg kg⁻¹ | mg kg⁻¹ | mg kg⁻¹ | mg kg⁻¹ | mg kg⁻¹ | mg kg⁻¹ | mg kg⁻¹ | mg kg⁻¹ | mM kg⁻¹ | g | kg m⁻³ |
| 50/50 | 6.53 | 93.0 | 5.2 | 1.9 | 7.0 | 505 | 61 | 596 | 92 | 4435 | 284 | 386 | 755 |
| 70/30 | 6.64 | 90.2 | 6.7 | 2.3 | 8.9 | 684 | 81 | 785 | 113 | 4741 | 319 | 311 | 606 |
| 100/0 | 6.47 | 86.1 | 8.9 | 2.9 | 11.8 | 953 | 111 | 1070 | 144 | 5200 | 372 | 163 | 259 |

extract liquid medium (ISP2) from the International Streptomyces Project [40] at 28 ˚C with agitation at 90 rpm for 10 d. Next, the culture was homogenized using sterile ceramic beads and the suspension was adjusted to 0.5 OD600 (concentration measured at 600 nm) in sterile physiological saline. The control treatment (C) was a substrate without inoculation with AMF or bacteria.

**Cultivation conditions.** Seeds were sterilized for 10 min in 3% sodium hypochlorite and washed with sterile distilled water. Sowing was conducted on April 2, 2020 directly into Teku V9 containers (square shape $9 \times 9$ cm, height 8 cm, volume 512 cm$^3$). Cultivation was performed in phytochamber Fytoscope 4400 (PSI, CZ) with the temperature at the germination stage of 23/20 ˚C (day/night). At 5 d, the cotyledons stage began and the temperature was reduced to 20/18 ˚C, followed by 22/19 ˚C. Relative air humidity was maintained at 85% for germination, followed 75% for the subsequent experimental period. Light intensity was set to 140 µmol m$^{-2}$ s$^{-1}$ for the germination stage, then increased to 200 µmol m$^{-2}$ s$^{-1}$ for the subsequent period of the trial, with 16 h of daylight. Irrigation with tap water was applied equally to all pots at a measured volume. Urea was used for fertilization (May 5, liquid 0.2% solution, 20 cm$^3$ per pot in irrigation doses), later the fertilizer YaraTera Kristalon 20 + 5 + 10 + 2 Azur was applied May 13 and 27 as a 0.1% liquid solution at a dose of 20 cm$^3$ per pot.

**Substrate sampling and analyses.** After the end of the experiment (June 3, 2020), samples of the substrate were taken to conduct laboratory analyses. A sample of approximately 150 g was taken from each replicate. The collected samples were air-dried, and basic parameters were determined, including pH (H$_2$O and KCl) using the potentiometric method, level of salinity using the conductometric method, and the capacity of the sorption complex by Kappen method. The content of total N and organic C was determined by elemental analysis using the Vario Max Cube apparatus (Elementar Analysensysteme GmbH, Langenselbold, Germany). Available forms of macroelements (Ca, Mg, Na, S, K, P) and microelements (Mn, Fe, Zn, Cu) were determined in the substrate samples by means of acetic acid after prior extraction according to the Nowosielski method [41]. The elements were determined by inductively coupled plasma atomic emission spectrometry using a Perkin Elmer Optima 7600 (PerkinElmer, US) spectrometer.

**Plant material sampling.** Tomato samples were collected on June 3, 2020, when all leaves were cut with scissors. Roots were completely extracted from the substrate and washed with distilled H$_2$O. Samples were stored immediately after harvest in a -80 ˚C deep freezer until analyses. For colonization analysis, root subsampling was conducted by selecting 10–20 mm randomly selected roots per container. These were fixed in a formaldehyde:ethanol:acetic acid 10%:50%:5% v/v solution (FAA) and stored in the dark at 4 ˚C before staining for microscopy [42].

Total aboveground fresh leaf weight and root fresh weight (FW) per plant was measured in five replicates per treatment.

**Staining and microscopy.** After fixation in FAA, roots were rinsed in distilled H$_2$O, then cleared in 2% KOH solution (1 h at 50 ˚C), and washed in distilled H$_2$O ($4 \times 3$ min). Roots were stained with a 0.03% (w/v) solution of Uvitex2B for 45 min at 90 ˚C, rinsed in distilled H$_2$O, and incubated in H$_2$O for 12 h. When the roots were placed on slides, a few drops of Hoechst/DAPI were added and covered with cover slip [43]. For staining with Alexa Fluor (AF) conjugates of wheat germ agglutinin (WGA), concanavalin A (Con A), and acid fuchsine, the tissues were fixed and cleared using the method described above. Roots were stained in a tube with a mixture consisting of WGA AF 594 conjugate (InvitrogenTM, USA) (50 µg mL$^{-1}$), Con A AF 647 (InvitrogenTM, USA) (50 µg mL$^{-1}$), and acid fuchsine (3%) at a ratio of 1:1:1, for 4–5 h at room temperature, rinsed in PBS ($4 \times 3$ min) and incubated for 12 h in PBS to remove all excess stain. Before mounting on the slide, few drops of Hoechst stain were added

to the slides with roots [44]. Mycorrhizal colonization quantification was conducted according to the grid-intersect method [45].

Confocal microscopy was completed using the LSM 800 (Carl Zeiss, Germany) microscope at 405/420–480 and 488/500–550 nm excitation/emission for Uvitex2B, 590/617 nm (excitation/emission) for WGA AF 594, 650/668 nm for Con A AF 647, and 350/461 nm for Hoechst stain. Lens used was 20x/0.8 NA. Processing of pictures was conducted in Zen Blue 3.0 (Carl Zeiss, Germany).

**Physiological parameters.** Plants at the stage of 30–40 cm height and having at least eight developed leaves were evaluated. Analysis of Normalized Difference Vegetation Index (NDVI) was performed on tomatoes before leaf harvest on May 25 by PlantPen model NDVI 310 (PSI, Ltd, CZ), a 50 mm$^2$ detector measuring at 635 and 750 nm. Seven fully developed leaves (the fourth from the plant apex) were analyzed per treatment. Chlorophyll absorbs visible light from 0.4 to 0.7 μm for photosynthesis, and cell structures strongly reflect near-infrared light from 0.7 to 1.1 μm. The differences in reflectance in the visible and near-infrared wavelengths were used to calculate the NDVI index.

On the same date (May 25), Quantum Yield (QY) was analyzed by the FluorPen model FP 110 (PSI Ltd, CZ) using seven tomato leaves per treatment (the same leaves as used for NDVI, but different leaf blade position). QY reflects photosystem II efficiency. In a 20 min dark-adapted leaf, this is equivalent to Fv/Fm.

**Analyses of stress biomarkers.** The antioxidant activity was determined in plant samples following DPPH radical (2,2-diphenyl-1-picrylhydrazyl) scavenging method [46]. The absorbance at λ = 517 nm of the mixture containing 0.1 cm$^3$ supernatant and 4.9 cm$^3$ 0.1 mM DPPH in 80% methanol was measured after 15 min of incubation in darkness at 20–22 ˚C with the UV-VIS Helios Beta spectrophotometer (Thermo Fisher Scientific, Inc., US). The antioxidant activity was calculated with the following formula: DPPH (%) = ((A0 − A1)/A0) × 100; where A0 is the absorbance of the reference solution and A1 is the absorbance of the test solution [47].

The total phenolics was estimated using the modified Folin-Ciocalteu colorimetric method [48]. A 2.5 g sample of leaves was mixed with 10 cm$^3$ of 80% methanol and centrifuged (3 492 × $g$, 15 min, 4 ˚C). The glass tubes were filled with 0.1 cm$^3$ of the supernatant and 2 cm$^3$ of sodium carbonate, left for 5 min, and then 0.1 cm$^3$ of Folin-Ciocalteu's reagent, mixed with deionized water (1:1 v/v) was added. After 45 min, phenols were determined by the colorimetric method at 750 nm using UV-VIS spectrophotometer against a reference solution. The total phenol value was expressed as gallic acid equivalents (mg GAE) per g FW.

To determine glutathione peroxidase (GPOX) activity, 2.5 g of leaves was grounded in an ice bath with 20 cm$^3$ of a 0.05 M potassium phosphate buffer and centrifuged (3 492 × $g$, 15 min, 4 ˚C). Then GPOX was assayed, according to Lück [49], with p-phenylenediamine as an electron donor and hydrogen peroxide as an oxidant. The reaction mixture contained the diluted supernatant, 0.05 M potassium phosphate buffer, p-phenylenediamine, and hydrogen peroxide. The absorbance at 485 nm was recorded at 60 s intervals for 2 min using a UV-VIS spectrophotometer. The GPOX activity was expressed in units (U) per g FW per min.

**Element concentration in plant tissues.** To determine the micro- and macroelement concentrations in the dry tomato samples, roots and leaves were mineralized in a mixture of $HNO_3$ solution and $H_2O_2$ at 1:3 (v:v). The weight of an analytical sample was max. 0.5 g dry weight (DW). Samples were acidified by adding 2 cm$^3$ $HNO_3$ per 100 cm$^3$ distilled water. The samples were concentrated 5-fold, and then the concentrations of elements (Ca, Mg, Na, S, K, P, Mn, Fe, Zn, and Cu) in the prepared solutions were determined by atomic emission spectroscopy (ICP) with an Optima 7600 spectrophotometer (Perkin Elmer, US), using the method described by Pasławski and Migaszewski [50].

**Statistical analysis.** The experiment was designed in a randomized complete block with seven replicates and two factors, representing different substrate compositions and different inocula used and the non-inoculated control. The substrate and plant samples for stress biomarkers and element concentrations were taken separately from every treatment and determined using three technical replicates. The samples for physiological parameters were measured using seven biological replications. The data were tested for normality of distribution according to the Shapiro and Wilk method [51] and homogeneity of variances using the Levene test [52]. A one- or two-way ANOVA was applied to test significance levels at $P \leq 0.05$ (*), $P \leq 0.01$ (**), or $P \leq 0.001$ (***) and non-significance (ns). Tukey's HSD (honest significant difference) test was used to separate means into homogenous groups. In the case of the one-way ANOVA, the experimental treatments were the source of variation. In the case of two-way ANOVA—plant parts (leaves and roots) were additional source of variation. The results were also examined using Pearson's correlation coefficient (r) between analyzed parameters. The principal component analysis (PCA), cluster (CA) analysis and heat maps were performed to precisely demonstrate and analyze the data and relationships among them. Correlations and PCA were used as supplementary statistical methods that enabled and expanded the analysis of the presented data and made additional relationships visible between experimental treatments and parameters. The results using raw data were presented for PCA analysis because no substantial differences appeared between raw and standardized data. All analyses were performed with the software Statistica 13.0 (Dell, Inc., USA).

## Results

### Substrate properties at the end of the experiment

The differences in the values characterizing the sorption complex in the substrate resulted from the share of the organic fraction in the substrates used in the experiment. The organic carbon content in the treatments C 100, AMF + AZ 100, and AMF + S 100 was approximately 50%, whereas, in the substrates of the remaining, this value ranged from approximately 3 to 7% of the DW of the substrate (Table 2). A statistically significant negative correlation

**Table 2. Effects of soil microorganisms on substrate physical and chemical characteristics after tomato transplants cultivation.**

| Treatment | Sum in the sorption complex (mM Na⁺ kg⁻¹) | | S + H (mM kg⁻¹) | Cation exchange capacity with alkaline cations (%) | Organic carbon (%) | Total nitrogen (%) | EC (mS) | pH | |
|---|---|---|---|---|---|---|---|---|---|
| | alkaline cations (S) | acid cations (H) | | | | | | H₂O | HCl |
| C 50* | 320 | 44 | 364 | 87.9 | 2.94 | 0.312 | 0.32 | 7.48 | 7.01 |
| C 70 | 512 | 56 | 568 | 90.1 | 6.82 | 0.649 | 1.45 | 7.08 | 6.70 |
| C 100 | 1352 | 172 | 1524 | 88.7 | 51.77 | 4.869 | 1.36 | 6.27 | 5.92 |
| AMF + AZ 50 | 396 | 52 | 448 | 88.4 | 2.91 | 0.303 | 0.99 | 7.06 | 6.86 |
| AMF + AZ 70 | 516 | 48 | 564 | 91.5 | 5.90 | 0.612 | 1.37 | 7.05 | 6.76 |
| AMF + AZ 100 | 1296 | 144 | 1440 | 90.0 | 49.18 | 4.682 | 1.48 | 6.55 | 6.17 |
| AMF + S 100 | 1272 | 148 | 1420 | 89.6 | 50.72 | 4.882 | 2.06 | 6.50 | 6.15 |

*C 50 –peat:sand ratio 50:50 (v:v) without inoculation; C 70 –peat:sand ratio 70:30 (v:v) without inoculation; C 100 –peat:sand ratio 100:0 (v:v) without inoculation; AMF + AZ 50 –peat:sand ratio 50:50 (v:v) inoculated with arbuscular mycorrhizal fungi (AMF) and Azospirillum brasilense (AZ), AMF + AZ 70 –peat: sand ratio 70:30 (v:v) inoculated with AMF and AZ, AMF + AZ 100 –peat:sand ratio 100:0 (v:v) inoculated with AMF and AZ; AMF + S– 100 peat:sand ratio 100:0 (v:v) inoculated with AMF and Saccharothrix tamanrassetensis (S).

occurred between the capacity of the sorption complex and the pH of the substrate (r = -0.985, $P \leq 0.01$) and between the capacity of the sorption complex and the content of organic carbon (r = -0.969, $P \leq 0.01$). A statistically significant positive correlation coefficient was found between the capacity of the sorption complex and salinity (r = 0.675, $P \leq 0.05$). In the case of the relationship between pH and salinity, the value of the correlation coefficient was r = -0.686, $P \leq 0.05$. The research results indicated different amounts of trace element forms available for plants in the substrates or treatment.

The greatest differentiation in the content of available forms of elements for plants was found for sodium (Na), phosphorus (P), and sulfur (S). The relative standard deviation for these elements was approximately 90%. The smallest variation occurred for calcium (Ca) and iron (Fe), where the variability in individual treatments was approximately 20%. The content of Ca and potassium (K) forms available to plants was similar, regardless of treatment. Potassium content in substrates after plant cultivation was in a range of 21 to 147 mg kg$^{-1}$ DW. In contrast, substrates composed of peat, with the addition of beneficial microorganisms (AMF + AZ 100 and AMF + S 100) contained the highest level of K after cultivation, and those composed of peat:sand in the ratio of 50:50 (v:v) contained the lowest (C 50, AMF + AZ 50) (Table 3). Calcium content in the substrate after tomato cultivation was the lowest in the control treatment composed of peat:sand at the ratio of 70:30 (v:v) (C 70), whereas it was significantly higher in the C 50 treatment and all treatments composed of peat (C 100, AMF + AZ 100, and AMF + S 100). In general, treatments composed of peat:sand in the ratio of 50:50 (v:v) and 70:30 (v:v), despite microorganism application, contained the lowest amounts of soluble S, K, P, magnesium (Mg), zinc (Zn), and copper (Cu) after tomato cultivation, which was opposite to that of substrates composed of peat (C 100, AMF + AZ 100, and AMF + S 100) (Table 3). The highest Cu contents were found in the C 100, AMF 100, and AMF + S 100 treatments. The content of this element in the substrates of these treatments was approximately 6-times higher than in the C 50 and AMF + AZ 50 treatments. A similar relationship was

**Table 3. Effects of soil microorganisms on soluble minerals (mg kg$^{-1}$ DW) in substrate after tomato transplants cultivation.**

| Treatment | Ca | Mg | Na | S | K |
|---|---|---|---|---|---|
| C 50* | 2619 ± 114 bc | 273 ± 11 a | 230 ± 14 a | 205 ± 15 a | 21 ± 0.8 a |
| C 70 | 1979 ± 51 a | 299 ± 12 a | 271 ± 15 a | 232 ± 55 a | 33 ± 7.1 a |
| C 100 | 3105 ± 170 d | 972 ± 62 c | 1614 ± 179 c | 1687 ± 168 c | 86 ± 5.4 bc |
| AMF + AZ 50 | 2362 ± 70 a-c | 255 ± 21 a | 206 ± 16 a | 209 ± 53 a | 33 ± 4.1 a |
| AMF + AZ 70 | 2272 ± 127 ab | 384 ± 28 a | 317 ± 26 a | 318 ± 22 a | 52 ± 4.0 ab |
| AMF + AZ 100 | 2768 ± 298 c | 865 ± 145 bc | 1159 ± 329 b | 1684 ± 40 c | 147 ± 35.0 d |
| AMF + S 100 | 2526 ± 92 bc | 740 ± 35 b | 852 ± 113 b | 1143 ± 116 b | 104 ± 10.3 c |
| | P | Mn | Fe | Zn | Cu |
| C 50 | 9.35 ± 0.4 a | 11.5 ± 0.18 d | 2.57 ± 0.17 b | 0.56 ± 0.01 a | 0.054 ± 0.01 a |
| C 70 | 15.63 ± 3.7 a | 8.42 ± 0.02 c | 2.48 ± 0.19 b | 0.51 ± 0.00 a | 0.087 ± 0.00 a |
| C 100 | 86.30 ± 12.8 c | 5.58 ± 0.23 b | 1.66 ± 0.02 a | 1.14 ± 0.06 b | 0.320 ± 0.03 c |
| AMF + AZ 50 | 11.42 ± 1.45 a | 8.51 ± 0.04 c | 1.74 ± 0.18 a | 0.57 ± 0.09 a | 0.046 ± 0.00 a |
| AMF + AZ 70 | 16.93 ± 0.4 a | 8.94 ± 0.19 c | 2.30 ± 0.03 b | 0.58 ± 0.05 a | 0.100 ± 0.01 a |
| AMF + AZ 100 | 70.19 ± 8.2 c | 5.03 ± 0.55 b | 2.27 ± 0.26 b | 0.95 ± 0.15 b | 0.291 ± 0.05 bc |
| AMF + S 100 | 46.69 ± 3.9 b | 4.31 ± 0.04 a | 1.74 ± 0.06 a | 0.90 ± 0.14 b | 0.247 ± 0.02 b |

*C 50 –peat:sand ratio 50:50 (v:v) without inoculation; C 70 –peat:sand ratio 70:30 (v:v) without inoculation; C 100 –peat:sand ratio 100:0 (v:v) without inoculation; AMF + AZ 50 –peat:sand ratio 50:50 (v:v) inoculated with arbuscular mycorrhizal fungi (AMF) and Azospirillum brasilense (AZ), AMF + AZ 70 –peat:sand ratio 70:30 (v:v) inoculated with AMF and AZ, AMF + AZ 100 –peat:sand ratio 100:0 (v:v) inoculated with AMF and AZ; AMF + S– 100 peat:sand ratio 100:0 (v:v) inoculated with AMF and Saccharothrix tamanrassetensis (S).

**Table 4. The value of the correlation coefficients between the individual substrate parameters after the end of the experiment.**

|      | Ca      | Mg       | Na       | S        | K        | P        | Mn       | Fe     | Zn       | Cu       |
|------|---------|----------|----------|----------|----------|----------|----------|--------|----------|----------|
| pH   | −0.656  | −0.972** | −0.950** | −0.950** | −0.832** | −0.956** | 0.923**  | 0.583  | −0.953** | −0.980** |
| Corg | 0.702*  | 0.972**  | 0.922**  | 0.964**  | 0.893**  | 0.933**  | −0.915** | −0.530 | 0.952**  | 0.979**  |

** Correlation coefficient significant at P ≤ 0.01.

found in the case of S and P. The substrate composed with peat:sand in the ratio of 50:50 (v:v) contained the highest amount of soluble manganese (Mn), whereas that formulated with peat enriched with beneficial microorganisms consortium (AMF + S 100) was characterized by the lowest Mn content after tomato cultivation. In the case of most of the examined elements, a statistically significant correlation occurred between the content of their assimilable forms in the media. The most common was a negative correlation between the reaction and the content of elements in the media and a positive correlation between organic C and the amount of assimilable forms of elements. The exceptions were Fe and Mn, where the relationships were opposite. The values of the correlation coefficients of most elements were approximately 0.9 (Table 4).

## Plant fresh weight

Fresh weight of leaves (mean 11.66 g) and roots (mean 3.44 g) was evaluated after extraction of samples from containers and exhibited significant treatment effects dependent upon the differences in the parameters (Fig 1). Although the lowest leaf and root FW values were recorded at C 50 and C 70, respectively, the highest leaf FW occurred in all treatments with peat media only (AMF + AZ 100, AMF + S 100, and C 100) showing the significantly positive impact of peat on plant growth. Treatment AMF + AZ 100 exhibited a significantly high root FW, together with AMF + S 100. The ratio of shoot/root weight was recorded in the range of 3.10 (C 70) to 3.95 (AMF + AZ 100) and leaf weight was correlated with root weight (r = 0.4781).

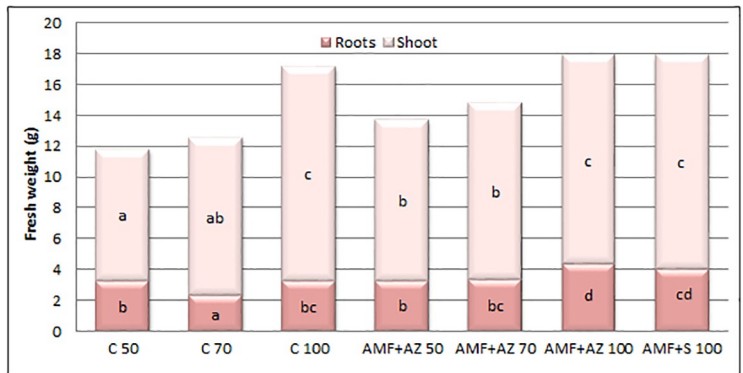

**Fig 1. Effects of soil microorganisms application on fresh weight of tomato shoots and leaves.** Means followed by different letters are significantly different at $p \leq 0.05$, with comparisons performed by Fisher's LSD test, separately for roots and shoots. C 50 –peat:sand ratio 50:50 (v:v) without inoculation; C 70 –peat:sand ratio 70:30 (v:v) without inoculation; C 100 ––peat:sand ratio 100:0 (v:v) without inoculation; AMF + AZ 50 –peat:sand ratio 50:50 (v:v) inoculated with arbuscular mycorrhizal fungi (AMF) and *Azospirillum brasilense* (AZ), AMF + AZ 70 ––peat:sand ratio 70:30 (v:v) inoculated with AMF and AZ, AMF + AZ 100 –peat:sand ratio 100:0 (v:v) inoculated with AMF and AZ; AMF + S–– 100 peat:sand ratio 100:0 (v:v) inoculated with AMF and *Saccharothrix tamanrassetensis* (S).

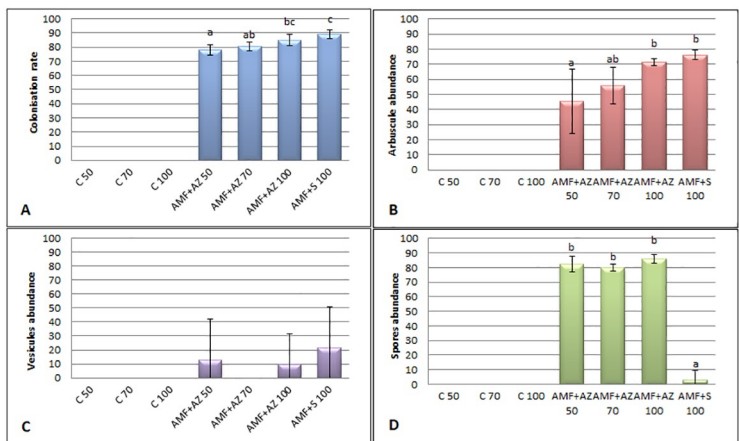

**Fig 2. Effects of soil microorganisms application mycorrhization parameters of tomato roots: Colonization rate (A), arbuscule abundance (B), vesicules abundance (C), and spores abundance (D).** Means followed by different letters are significantly different at $P \leq 0.05$, with comparisons performed by Fisher's LSD test, only for mycorrhized objects. C 50—peat:sand ratio 50:50 (v:v) without inoculation; C 70 –peat:sand ratio 70:30 (v:v) without inoculation; C 100 –peat:sand ratio 100:0 (v:v) without inoculation; AMF +AZ 50 –peat:sand ratio 50:50 (v:v) inoculated with arbuscular mycorrhizal fungi (AMF) and *Azospirillum brasilense* (AZ), AMF + AZ 70 –peat:sand ratio 70:30 (v:v) inoculated with AMF and AZ, AMF + AZ 100 –peat:sand ratio 100:0 (v:v) inoculated with AMF and AZ; AMF + S– 100 peat:sand ratio 100:0 (v:v) inoculated with AMF and *Saccharothrix tamanrassetensis* (S).

## Root colonization

Results of confocal microscopy confirmed successful root colonization of mycorrhizal fungi in all treatments inoculated by AMF. Data in Fig 2A show the high mean level for root colonization rate in AMF + AZ 50, AMF + AZ 70, AMF + AZ 100, and AMF + S 100 in the range from 78 to 89%. Root colonization in the control treatment was not detected.

Following root colonization rate, arbuscular abundance was analyzed. As Fig 2B shows, the arbuscules were present from 45 to 76%. These levels indicated the intensive process of mutual symbiosis. Mycorrhizal fungi also developed spores detected by microscopic observation (Fig 2D).

Their presence occurred in AMF + AZ 50, AMF + AZ 70, and AMF + AZ 100, but not in AMF + S.

Supporting figures from microscopic observation demonstrate results. S1 and S2 Figs show the interaction of mycorrhizal fungi in root tissue and bacterial colonies in the root hair area. Root samples carefully extracted from the substrate revealed the development of dense mycelial structures surrounding root hairs of tomatoes (S1_1 Fig). Conditions with higher organic matter content led to more abundant grid structures of fungi on roots. In most of the AMF treatments, the development of spores was detected. This situation was also found in the low organic matter content treatment (AMF + AZ 50) and confirmed the ability of AMF to colonize plant roots and form propagative structures (S1_2 Fig).

S1_3A and S1_3B Fig show effective symbiosis between mycorrhizal fungi and tomato plants. Arbuscular structures in root tissues correspond to positive interactions of host and fungi in water/nutrient exchange. Treatments with 100% peat showed higher levels of arbuscules abundance, although symbiosis functioned in the peat:sand substrate (50:50) as well.

Treatments containing the bacterial inoculant *Azospirillum brasilense* were observed for the determination of bacterial colonies on roots. In all treatments with this inoculation, the bacterial colonies were abundant, as shown by the specific probe for *A. brasilense* (S2 and S3 Figs).

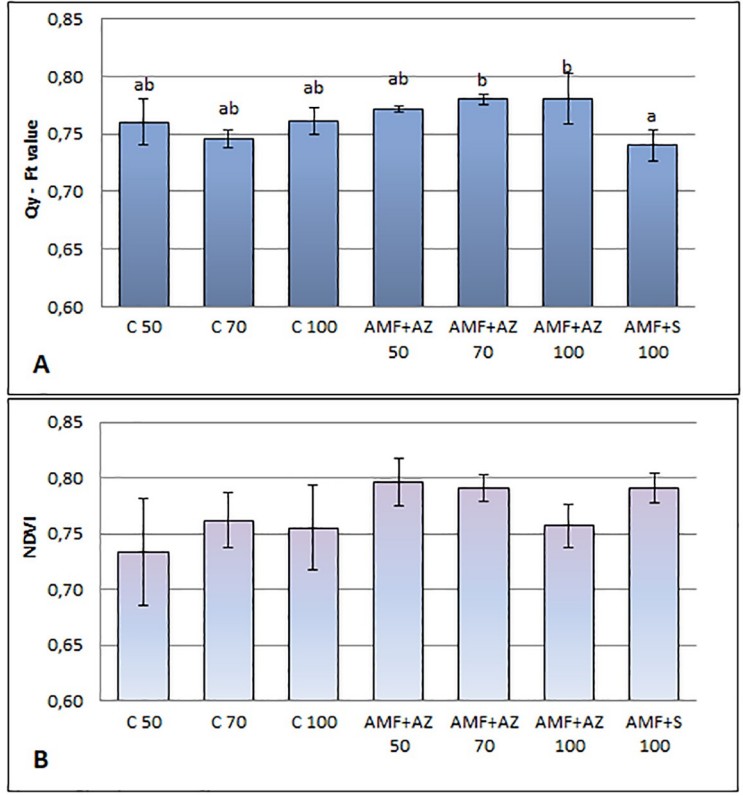

**Fig 3. Effects of soil microorganisms application to substrates of different organic matter content on Quantum Yield (QY) (A) and Normalized Difference Vegetation Index (NDVI) (B) in tomato leaves transplants.** Means followed by different letters are significantly different at $P \leq 0.05$, with comparisons performed by Fisher's LSD test. Error bars represent the standard deviation (± SD) for interaction. C 50 –peat:sand ratio 50:50 (v:v) without inoculation; C 70 –peat:sand ratio 70:30 (v:v) without inoculation; C 100 –peat:sand ratio 100:0 (v:v) without inoculation; AMF + AZ 50 –peat:sand ratio 50:50 (v:v) inoculated with arbuscular mycorrhizal fungi (AMF) and *Azospirillum brasilense* (AZ), AMF + AZ 70 –peat:sand ratio 70:30 (v:v) inoculated with AMF and AZ, AMF + AZ 100 –peat:sand ratio 100:0 (v:v) inoculated with AMF and AZ; AMF + S– 100 peat:sand ratio 100:0 (v:v) inoculated with AMF and *Saccharothrix tamanrassetensis* (S).

## Physiological parameters

The NDVI as a parameter corresponding to the physiological status of plants was 0.77 (mean for all treatments), according to the evaluation of tomato leaf fluorescence activity. The higher NDVI levels occurred in AMF + AZ 50,70 and AMF + S. The NDVI of control treatments was lower, especially in C 50, but differences were not significant statistically (Fig 3).

Quantum yield expressed by the Ft value exhibited a non-significant correlation with NDVI, where Qy = 0.76718–0.0055*NDVI corresponding to the tight link among both physiological parameters (Fig 4). There was no significant difference in this value among control treatments. The values ranged from 70 AMF + AZ and 100 to the AMF + S with the lowest value.

## Analyses of stress biomarkers

The antioxidant activity was considered in roots and shoots of tomato seedlings affected by different growth conditions followed by physiological acclimatization processes. Generally, the antioxidant activity measured as DPPH scavenging activity was approximately 2-times higher in tomato leaves than in roots for each treatment, except AMF + AZ 100 (Fig 5A). In general,

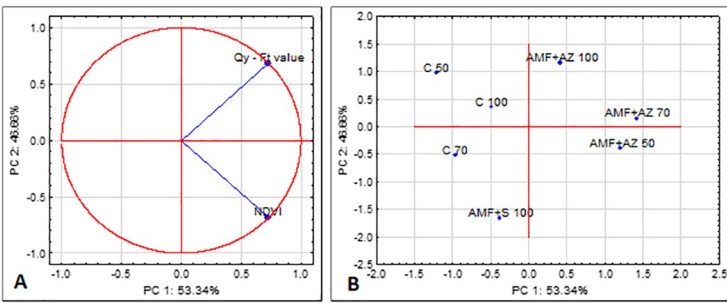

**Fig 4. Bi-plot presenting the correlation between the tested fluorescence parameters of tomato transplant leaves (A) and ordination illustrating differences between substrates tested towards the fluorescence parameters of tomato transplant leaves (B).** C 50 –peat:sand ratio 50:50 (v:v) without inoculation; C 70 –peat:sand ratio 70:30 (v:v) without inoculation; C 100 –peat:sand ratio 100:0 (v:v) without inoculation; AMF + AZ 50 –peat:sand ratio 50:50 (v:v) inoculated with arbuscular mycorrhizal fungi (AMF) and *Azospirillum brasilense* (AZ), AMF + AZ 70 –peat:sand ratio 70:30 (v:v) inoculated with AMF and AZ, AMF + AZ 100 –peat:sand ratio 100:0 (v:v) inoculated with AMF and AZ; AMF + S– 100 peat:sand ratio 100:0 (v:v) inoculated with AMF and *Saccharothrix tamanrassetensis* (S).

AM fungi and *Azospirillum brasilense* (AMF + AZ) applied to the substrate reduced tomato seedling antioxidant activity. Tomato seedlings, grown in the substrate composed of peat and inoculated with AMF and *Saccharothrix tamanrassetensis* (AMF + S), showed the highest antioxidant activity. The roots of plants cultivated in a substrate composed of peat:sand at the ratio 50:50 (v:v) and inoculated with AMF + AZ. The lowest DPPH scavenging activity in roots was recorded in treatment AMF + AZ in substrate with 50:50 peat:sand ratio, while the highest DPPH activity in the leaves at AMF + S treatment was found. Antioxidant activity of tomato leaf extracts was positively correlated with total phenolics in leaves (Table 1), a similar relation was observed for roots. These dependences are illustrated (Fig 6A) by the distance of the eigenvectors of antioxidant activity and total phenolics and narrow angles between the eigenvectors.

The total phenolics (TP) content was significantly higher in tomato seedling leaves than roots, considering the main effects analysis (Fig 5B). Tomato roots contained a similar phenolics level, and significant differences were noted for C 70, AMF + AZ 50, and AMF + AZ 70 treatments, whereas the AMF + S 100 treatment had a higher level. Analysis of the TP in tomato leaves shoved its higher level in samples from plants of C 50 and C 70 treatments, without microbiota inoculation. Total phenolics in tomato leaves were positively correlated with Ca Mg, Na, and Mn contents in leaves and roots, Fe in roots, and Cu in leaves. The correlations of TP and mineral elements in tomato roots were positive for S and Zn, whereas a negative relationship was noted for Ca, Mn, and Fe. The correlations of TP in tomato roots and mineral elements in tomato leaves were positive only for Fe, but negative for K, Ca, Mg, Na, and Mn (Table 2).

The tomato seedling roots showed a higher GPOX activity than leaves, except for the plants grown in peat:sand at a ratio of 50:50 (v:v), without microbiota inoculation (Fig 5C). The most notable differences were for roots (the lowest GPOX activity) and the roots (the highest GPOX activity) of tomato seedlings sampled from the substrate with peat:sand ratio 70:30 (v:v), inoculated with AMF + AZ. GPOX activity in roots was 2-times higher than in leaves of the seedlings of this treatment. The conversely directed eigenvectors of GPOX activity and total phenolics in roots of tomato seedlings in Fig 4a illustrated the negative correlation between these parameters, with a correlation coefficient of r = -0.514, $P \leq 0.05$. However, the correlation between the aforementioned parameters in tomato leaves was positive. Correlation coefficients between GPOX activity of tomato seedlings roots and shoots are presented in Table 2.

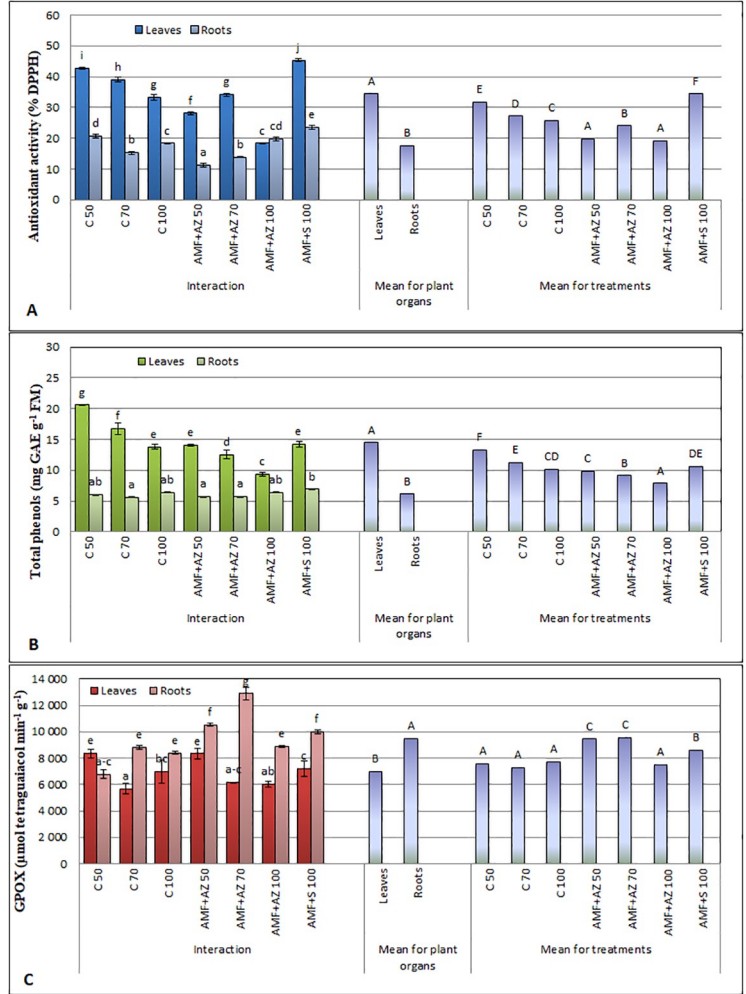

**Fig 5. Effects of soil microorganisms application to substrates of different organic matter content on antioxidant activity (A), total phenols (B), and glutathione peroxidase (GPOX) (C) activity in leaves and roots of tomato transplants.** Means followed by different letters are significantly different at $P \le 0.05$, with comparisons performed by Fisher's LSD test. Error bars represent the standard deviation (± SD) for interaction. C 50 –peat:sand ratio 50:50 (v:v) without inoculation; C 70 –peat:sand ratio 70:30 (v:v) without inoculation; C 100 –peat:sand ratio 100:0 (v:v) without inoculation; AMF + AZ 50 –peat:sand ratio 50:50 (v:v) inoculated with arbuscular mycorrhizal fungi (AMF) and *Azospirillum brasilense* (AZ), AMF + AZ 70 –peat: sand ratio 70:30 (v:v) inoculated with AMF and AZ, AMF + AZ 100 –peat:sand ratio 100:0 (v:v) inoculated with AMF and AZ; AMF + S– 100 peat:sand ratio 100:0 (v:v) inoculated with AMF and *Saccharothrix tamanrassetensis* (S).

PCA analysis illustrated that the C 50 treatment contributed significantly and negatively to PC1 and PC2, whereas AMF + AZ 100 contributed positively to PC1 and PC2 (Fig 6B).

## Elements concentration in plant tissues

The mineral content in substrates and tomato seedlings was significantly affected by substrate composition and microbial inoculation. The lowest potassium root:shoot ratio occurred in plants collected from C 50 and AMF + AZ 50 treatments and was equal to 0.56 and 0.55, respectively, whereas the highest was determined for plants in the C 70 substrate (0.82) (Fig 7). Potassium was accumulated in tomato leaves in significantly higher amounts than in the roots, which was visually confirmed with the use of heat maps (Fig 8A). Leaves of plants collected

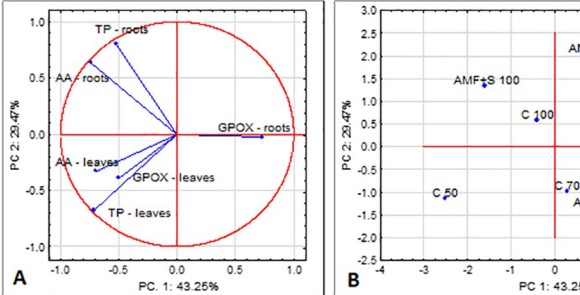

**Fig 6. Bi-plot presenting the correlation between the tested antioxidant parameters of tomato transplants (A) and ordination illustrating differences between substrates tested towards the antioxidant parameters of tomato transplants (B).** C 50 –peat:sand ratio 50:50 (v:v) without inoculation; C 70 –peat:sand ratio 70:30 (v:v) without inoculation; C 100 –peat:sand ratio 100:0 (v:v) without inoculation; AMF + AZ 50 –peat:sand ratio 50:50 (v:v) inoculated with arbuscular mycorrhizal fungi (AMF) and *Azospirillum brasilense* (AZ), AMF + AZ 70 –peat:sand ratio 70:30 (v:v) inoculated with AMF and AZ, AMF + AZ 100 –peat:sand ratio 100:0 (v:v) inoculated with AMF and AZ; AMF + S– 100 peat:sand ratio 100:0 (v:v) inoculated with AMF and *Saccharothrix tamanrassetensis* (S).

from AMF + AZ 50 and AMF + AZ 70 treatments contained the highest total K level, whereas roots of plants grown in the C 50 treatment had the lowest (Table 4). The correlation coefficients between elements in plants and soil are presented in Table 1.

The leaves sampled from plants grown in C 50, C 70, AMF + AZ 50, and AMF + AZ 70 treatments contained the highest amounts of Ca, whereas the lowest was determined in plant roots in AMF + AZ 100 and AMF + S 100 treatments (Table 4, Fig 8B). The Ca root:shoot ratio was 0.30 for the C 70 treatment (the lowest value) and 0.43 for the AMF + AZ 100 treatment (the highest value) (Fig 7).

The distribution of Mg was different among plant parts and treatments, and the root:shoot ratios were in the range from 0.69 (C 70) to 1.27 (AMF + AZ 100) (Fig 7). Roots of tomato seedlings from the control treatments contained higher Mg levels than leaves, whereas an inverse relationship was noted for the AMF + AZ 70 treatment. In the remaining treatments, differences between roots and leaves concerning Mg content were not significant. The highest

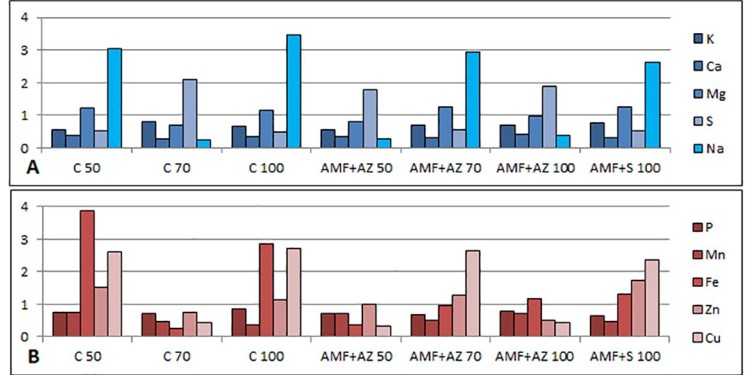

**Fig 7. Effects of soil microorganisms application to substrates of different organic matter content on root: Shoot ratio of mineral elements in tomato transplants.** C 50 –peat:sand ratio 50:50 (v:v) without inoculation; C 70 –peat: sand ratio 70:30 (v:v) without inoculation; C 100 –peat:sand ratio 100:0 (v:v) without inoculation; AMF + AZ 50 –peat: sand ratio 50:50 (v:v) inoculated with arbuscular mycorrhizal fungi (AMF) and *Azospirillum brasilense* (AZ), AMF + AZ 70 –peat:sand ratio 70:30 (v:v) inoculated with AMF and AZ, AMF + AZ 100 –peat:sand ratio 100:0 (v:v) inoculated with AMF and AZ; AMF + S– 100 peat:sand ratio 100:0 (v:v) inoculated with AMF and *Saccharothrix tamanrassetensis* (S).

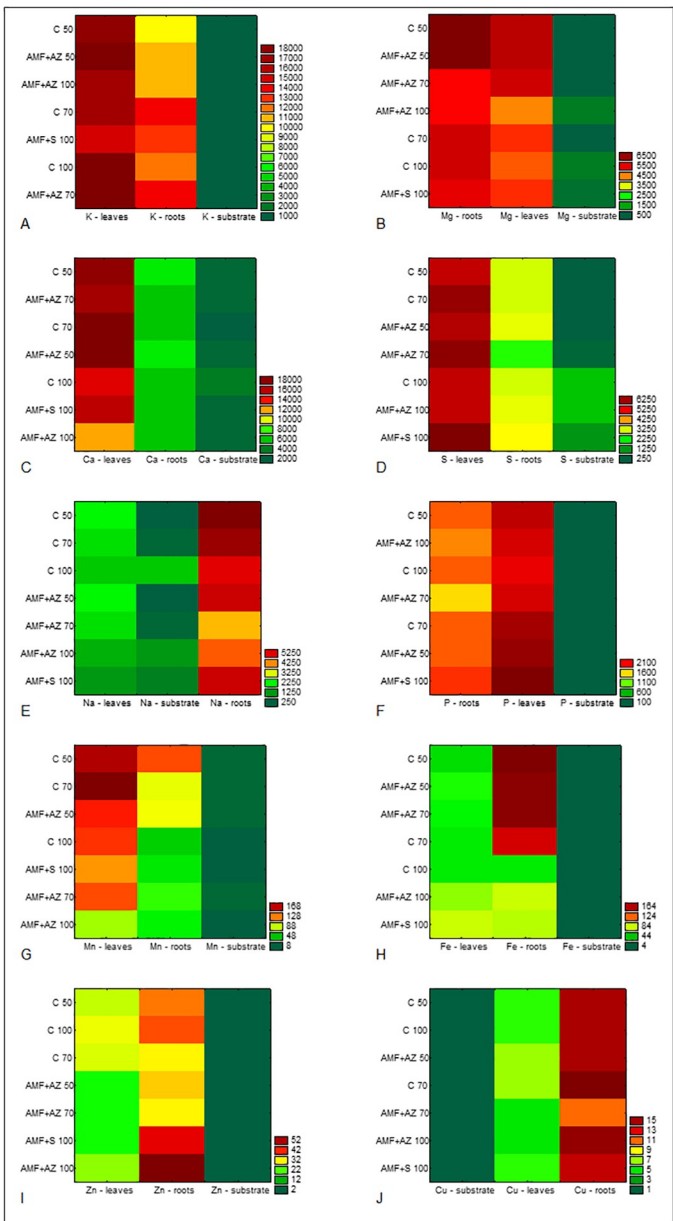

**Fig 8. Heat map plots of elements content in soil/plant system under different substrate treatments: Potassium (A), magnesium (B), calcium (C), sulfur (D), sodium (E), phosphorus (F), manganese (G), iron (H), zinc (I), and copper (J).** Green colors indicate that element contents were less than the means, while red colors indicate that element contents were higher than the means. C 50 –peat:sand ratio 50:50 (v:v) without inoculation; C 70 –peat:sand ratio 70:30 (v:v) without inoculation; C 100 –peat:sand ratio 100:0 (v:v) without inoculation; AMF + AZ 50 –peat:sand ratio 50:50 (v:v) inoculated with arbuscular mycorrhizal fungi (AMF) and *Azospirillum brasilense* (AZ), AMF + AZ 70 –peat:sand ratio 70:30 (v:v) inoculated with AMF and AZ, AMF + AZ 100 –peat:sand ratio 100:0 (v:v) inoculated with AMF and AZ; AMF + S– 100 peat:sand ratio 100:0 (v:v) inoculated with AMF and *Saccharothrix tamanrassetensis* (S).

Mg amount was in roots of plants grown in C 50 and AMF + AZ 50 substrates (Table 4, Fig 8C).

The sulfur root:shoot ratio was the highest in plants collected from the C 70 treatment (2.09) followed by the AMF + AZ 100 and AMF + AZ 50 (1.89 and 1.80, respectively) (Fig 7). Tomato leaves collected from all substrates contained a higher amount of sulfur than roots,

**Table 5. Effects of soil microorganisms application to substrates of different organic matter content on potassium, calcium, manganese, sulphur, and sodium content in leaves and roots of tomato transplants.**

| Treatment | Plant part | K | Ca | Mg | S | Na |
|---|---|---|---|---|---|---|
| | | (mg kg$^{-1}$ DW) | | | | |
| C 50* | Leaves | 17413 ± 462 fg** | 17752 ± 94 ef | 5825 f-g | 5566 ± 95 c | 2228 ± 63 d |
| | Roots | 9725 ± 346 a | 6806 ± 112 b | 7180 h | 3027 ± 274 b | 6822 ± 40 h |
| C 70 | Leaves | 16206 ± 436 ef | 18436 ± 215 f | 4952 a-d | 6331 ± 217 de | 1787 ± 35 bc |
| | Roots | 13337 ± 296 cd | 5500 ± 393 ab | 5712 e-g | 3182 ± 393 b | 6199 ± 119 g |
| C 100 | Leaves | 18245 ± 2120 fg | 14532 ± 406 d | 4537 ab | 5735 ± 406 c | 1737 ± 109 bc |
| | Roots | 11935 ± 985 a-c | 5359 ± 443 a | 5655 c-g | 3154 ± 443 b | 5134 ± 424 f |
| AMF + AZ 50 | Leaves | 18646 ± 602 g | 18720 ± 767 f | 5626 e-h | 5974 ± 769 c | 2028 ± 35 cd |
| | Roots | 10267 ± 859 ab | 6358 ± 58 ab | 7168 h | 3251 ± 71 b | 5297 ± 71 f |
| AMF + AZ 70 | Leaves | 18891 ± 292 g | 16609 ± 1355 e | 5958 g | 6509 ± 1357 e | 1834 ± 150 b-d |
| | Roots | 13216 ± 987 cd | 5563 ± 85 ab | 5138 b-e | 2440 ± 85 a | 3955 ± 29 f |
| AMF + AZ 100 | Leaves | 14635 ± 234 de | 11865 ± 97 c | 4293 a | 5656 ± 97 c | 1489 ± 80 ab |
| | Roots | 10242 ± 769 ab | 5122 ± 324 a | 5021 a-e | 3293 ± 324 b | 4318 ± 106 f |
| AMF + S 100 | Leaves | 16075 ± 234 ef | 15050 ± 173 d | 4872 a-c | 7029 ± 173 f | 1156 ± 90.0 a |
| | Roots | 12378 ± 570 b-d | 5025 ± 286 a | 5470 c-g | 3513 ± 286 b | 5305 ± 23.0 f |
| Source of variation | | | | | | |
| Treatment (T) | | *** | *** | *** | *** | *** |
| Plant part (P) | | *** | *** | *** | *** | *** |
| T×P | | *** | *** | *** | *** | *** |

*C 50 –peat:sand ratio 50:50 (v:v) without inoculation; C 70 –peat:sand ratio 70:30 (v:v) without inoculation; C 100 –peat:sand ratio 100:0 (v:v) without inoculation; AMF + AZ 50 –peat:sand ratio 50:50 (v:v) inoculated with arbuscular mycorrhizal fungi (AMF) and Azospirillum brasilense (AZ), AMF + AZ 70 –peat:sand ratio 70:30 (v:v) inoculated with AMF and AZ, AMF + AZ 100 –peat:sand ratio 100:0 (v:v) inoculated with AMF and AZ; AMF + S– 100 peat:sand ratio 100:0 (v:v) inoculated with AMF and Saccharothrix tamanrassetensis (S).

**Means within a column, followed by different letters are significantly different, with comparisons performed using Tukey's HSD test. Levels of significance:

*** P ≤ 0.001.

especially those sampled from the AMF + AZ 70 and AMF + S 100 treatments (Table 4, Fig 8D).

The root:shoot accumulation ratios were 0.26, 0.28, and 0.40 for C 70, AMF + AZ 50, and AMF + AZ 100, respectively, whereas its value was close to 3 for the remaining treatments (Fig 7). Tomato seedlings accumulated Na in roots, especially those from the C 70 treatment (Table 4, Fig 8E).

P accumulated in leaves, especially in seedlings collected from the AMF + S 100 treatment, followed by the AMF + AZ 50 and C 70 treatments. The P content was lower in the roots than leaves. Moreover, differences between treatments were not statistically significant, with the root:shoot ratio below 1 (Table 5, Figs 7 and 8F).

The leaf samples collected from the C 70 and C 50 treatments contained the highest P amount, as well as the root samples collected from the AMF + AZ 50 and C 70 treatments (Table 5, Fig 7). Similar to P, Mn content was higher in tomato leaves than roots, with the root: shoot accumulation ratio in the range from 0.37 (C 100) to 0.76 (C 50) (Fig 8F).

The substrates of control treatments, composed of peat:sand at the ratio 50:50 (v:v) and 70:50 (v:v), as well as the AMF + AZ 70 and AMF + AZ 100 treatments, contained significantly higher levels of soluble Fe compared to the remaining treatments (Table 3). Fe content was the highest in tomato seedling roots sampled in the treatments C 50, AMF + AZ 100, and AMF + AZ 70, followed by C 70. In general, roots and leaves of tomatoes collected from the remaining

**Table 6. Effects of soil microorganisms application to substrates of different organic matter on phosphorus, manganese, iron, zinc, and copper content in leaves and roots of tomato transplants.**

| Treatment | Plant part | P | Mn | Fe | Zn | Cu |
|---|---|---|---|---|---|---|
| | | (mg kg⁻¹ DW) | | | | |
| C 50* | Leaves | 2382 ± 99 e–g** | 174.29 ± 3.51 h | 50.49 ± 0.31 a | 24.57 ± 1.11 bc | 5.65 ± 0.02 a |
| | Roots | 1826 ± 101 a–c | 132.42 ± 1.93 fg | 194.86 ± 9.14 f | 37.05 ± 1.29 f | 14.66 ± 0.43 cd |
| C 70 | Leaves | 2592 ± 14 f–h | 206.46 ± 3.89 i | 53.32 ± 1.42 ab | 27.17 ± 1.13 cd | 6.29 ± 0.13 a |
| | Roots | 1876 ± 213 a–d | 99.21 ± 1.11 e | 151.68 ± 19.8 e | 30.61 ± 1.98 de | 16.97 ± 0.20 e |
| C 100 | Leaves | 2146 ± 106 c–e | 137.24 ± 6.83 g | 54.38 ± 2.32 ab | 29.97 ± 1.62 de | 5.64 ± 1.13 a |
| | Roots | 1851 ± 153 a–c | 50.46 ± 4.17 a | 52.62 ± 4.34 ab | 38.33 ± 3.16 f | 14.83 ± 1.22 cd |
| AMF + AZ 50 | Leaves | 2652 ± 95 gh | 142.01 ± 5.48 g | 62.05 ± 1.24 b–d | 19.57 ± 1.30 a | 6.28 ± 0.13 a |
| | Roots | 1886 ± 7 a–d | 102.84 ± 1.80 e | 82.01 ± 2.12 d | 33.8 ± 1.737 ef | 14.79 ± 0.41 cd |
| AMF + AZ 70 | Leaves | 2225 ± 240 de | 134.73 ± 0.74 g | 58.68 ± 2.03 a–c | 18.20 ± 0.60 a | 4.76 ± 0.22 a |
| | Roots | 1529 ± 12 a | 69.31 ± 2.21 c | 187.31 ± 7.64 f | 31.15 ± 1.92 de | 10.47 ± 0.74 b |
| AMF + AZ 100 | Leaves | 2279 ± 13 e–g | 83.61 ± 3.24 d | 72.38 ± 0.92 b–d | 22.33 ± 1.74 ab | 4.64 ± 0.14 a |
| | Roots | 1782 ± 63 ab | 60.55 ± 1.37 bc | 187.84 ± 5.58 f | 60.66 ± 0.90 h | 15.19 ± 0.66 d |
| AMF + S 100 | Leaves | 2934 ± 17 h | 123.25 ± 0.26 f | 80.43 ± 0.26 d | 19.69 ± 0.20 a | 5.22 ± 0.17 a |
| | Roots | 1917 ± 174 b–d | 58.37 ± 2.02 ab | 78.44 ± 4.65 cd | 44.21 ± 1.30 g | 13.42 ± 0.35 c |
| Source of variation | | | | | | |
| Treatment (T) | | *** | *** | *** | *** | *** |
| Plant part (P) | | *** | *** | *** | *** | *** |
| T×P | | *** | *** | *** | *** | *** |

*C 50 –peat:sand ratio 50:50 (v:v) without inoculation; C 70 –peat:sand ratio 70:30 (v:v) without inoculation; C 100 –peat:sand ratio 100:0 (v:v) without inoculation; AMF + AZ 50 –peat:sand ratio 50:50 (v:v) inoculated with arbuscular mycorrhizal fungi (AMF) and Azospirillum brasilense (AZ), AMF + AZ 70 –peat:sand ratio 70:30 (v:v) inoculated with AMF and AZ, AMF + AZ 100 –peat:sand ratio 100:0 (v:v) inoculated with AMF and AZ; AMF + S– 100 peat:sand ratio 100:0 (v:v) inoculated with AMF and Saccharothrix tamanrassetensis (S).

**Means within a column, followed by different letters are significantly different, with comparisons performed using Tukey's HSD test. Levels of significance:

*** $P \leq 0.001$.

substrates contained significantly lower amounts of Fe, which were primarily not differentiated between treatments or plant parts (Table 6, Fig 7). The root:shoot Fe ratio was also distinguished among treatments and was in the range from 0.27 (C 70) to 3.86 (C 50) (Fig 8H).

Roots of plants grown in the AMF + AZ 100 substrate contained the highest level of Zn, followed by AMF + S 100, with a root:shoot ratio of 0.51 and 1.72, respectively. Analysis of the chemical composition of leaves shoved significantly higher Zn content in leaves of plants collected from the treatments C 50 –C 70 compared to those collected from the microbiota application treatments (Table 6, Figs 7 and 8I).

Leaves of tomatoes from all treatments did not differ in Cu content, which was generally lower than that found in roots. Different concentrations of Cu characterized roots of investigated treatments, with the highest values in plant roots from the C 70 treatment and the lowest from the AMF + AZ 100 treatment (Table 6, Fig 8J). The root:shoot accumulation ratio was 0.33, 0.42, and 0.43 for plants cultivated in the substrate AMF + AZ 50, and AMF + AZ 100, and C 70, respectively, whereas its value was close to 2.5 for the remaining treatments (Fig 7).

## Discussion

### Substrate characteristics

Substrate analysis after tomato cultivation in the present research confirmed the crucial significance of organic matter content for its physical and chemical characteristics. The highest total

N, C-organic, sum of alkaline cations, and salinity, and the lowest pH was in substrates composed of peat, and the inoculation with AMF and bacteria had slight effect on these parameters. The substrates exhibited significantly different mineral composition after tomato cultivation. Even under conditions of high salinity, the mineral soluble forms were not reduced, especially in inoculated treatments. The uptake of N, P, Mg, Ca, Mn, and Fe were enhanced in inoculated tomato [53]. The chemical properties of the substrate influenced the amount of biomass produced [54,55].

The substrate composition affects the bioavailability of elements, both necessary and potentially toxic for plants. The capacity of the sorption complex plays a key role in the release of ions into the soil solution, associated with changes in the salinity level and soil pH. The capacity of the sorption complex in the tested substrates ranged from 364 to 1524 mM kg$^{-1}$, and was proportional to the share of the organic component in the substrate. The lowest values were found in treatments C 50, C 70, AMF + AZ 50, and AMF + AZ 70, while the highest values in all treatments with peat only. Despite significant differences in this parameter from individual treatments, it was not reflected in the level of saturation of the sorption complex with alkaline cations, which ranged from 87.9% to 91.5%. Under conditions of maintaining the optimal fertilization and irrigation, the capacity of the sorption complex does not influence the development of plants [56]. A more important parameter is the saturation of the sorption complex with alkaline cations, which indicates the potential negative impact of hydrogen ions on plant growth. The pH reaction is related to the level of saturation of the sorption complex. Despite the large differences in the content of organic fraction in the substrates and the capacity of the sorption complex, slight differences in the substrate pH occurred after the experiment. No translation was found between the amount of organic materials in the substrate and its strategic properties. This could be caused by the short duration of the experiment. The content of elements was low and in the treatments with a small amount of organic component in the substrate, they could be assessed as deficient for plants [54]. This is especially true of P and K. The content of micronutrients and potentially toxic elements in no case indicated a threat to plants [57]. A negative correlation was found between the reaction and the content of most elements in the medium, and a positive correlation between the content of organic C and the amount of assimilable forms of the elements. The exceptions were Fe and Mn with opposite relationships.

## Plant weight

Shoot and root DW and leaf area of tomatoes grown in a low P soil-sand mix were higher in mycorrhizal than in nonmycorrhizal plants [58]. Ribaudo et al. [59] determined that tomato inoculation with *Azospirillum brasilense* FT326 significantly enhanced the root and shoot weight. Increase in biomass weight was correlated with increased absorption of mineral elements, such as N, P, K, Ca, and Mg which was noticed under inoculation with single and combined bacterial strains, i.e., *Azotobacter* or *Azospirillum*. Similar dependence was found in the present research.

## Root colonization

The mycorrhizosphere and sporosphere bacteria can boos germination and/or improving the growth of extraradical mycelium, the fine absorbing network of hyphae extend around the roots [60]. Colonization was confirmed in all treatments inoculated by AMF or *A. brasilense*. The colonization rate was increased in peat (100%), which showed a higher ability of AMF to develop symbiosis, contrary to the sand. Paranavithana et al. [61] found the higher C soil content corresponded to the higher AMF colonization in rice. The results showed the same correlation in the rate of colonization and C content. AMF colonizing plants could support

rhizodeposition in soil, and as a result mycorrhiza increased plant C sequestration [62]. This effect improves soil amelioration of degraded soils by higher organic matter deposition and sorption of water and nutrients.

*Saccharothrix* spp. is producing dithiolopyrolone derivates with antifungal activity [21]. Such studies were background for our goal of confirming whether *Saccharothrix* could negatively affect formation of mycorrhizal symbiosis. Results have shown good development of AMF in co-inoculation with *S. tamanrassetensis* and other positive effects on some others analytical parameters.

## Physiological parameters

NDVI relates to the chlorophyll content in leaves and provides information on the photosynthetic activity [63,64]. The positive impact of microorganism was not documented in our study. However, this data was in line with Nogales et al. [65] results on *F. mosseae*-inoculated grapevine plants, which showed the decrease in NDVI after 3 months growth in Cu-contaminated soil. The Cu-contamination created stressful soil conditions, which could develop a similar chain of basic physiological reactions. The quantum yield of photosynthesis is a widely used measurement in many studies [66]. Quantum yield and NDVI values represent better nitrogen plant management. Many studies [67] have shown a positive correlation to the increased N fertilization in tomatoes. However, we did not confirm this effect in AMF or AZ treatments. Although in AMF it could be expected because most AMF are phosphorus plant uptake enhancers, *Azospirillum* is a typical N-fixing bacterium. The possibility of quantitative and time-dependent limitation of microbial colony formation [68] in mutual AMF and *Azospirillum* presence in the substrate could lead to this result.

## Analyses of stress biomarkers

Plants grown in substrate reflecting degraded soil conditions are under stress conditions modifying root system growth, functioning, and mineral absorption effectiveness. PGPM can partially compensate the chemical fertilizers, especially in tomatoes, a highly mycorrhizal-dependent crop [69]. The present study showed the new aspect of this relationship because tomato grown in substrate with AMF and *Azospirillum brasilense* showed the lowest antioxidant activity. In contrast, those inoculated with AMF and *S. tamanrassetensis* showed the highest antioxidant activity in both roots and leaves. The present study confirmed the organ-dependent polyphenol concentration. Inculet et al. [70] showed long-term effects of PGPM inoculation on tomato growth, yield, and fruit polyphenol content and antioxidant activity. In general, plants exhibit an increased synthesis of polyphenols under abiotic stress, including suboptimal soil conditions [71]. Abiotic stresses create osmotic stress [72], oxidative damage [73], and reactive oxygen species (ROS) [74], that lead to numerous physio-molecular changes, including a decrease in photosynthetic activities [75], DNA, protein and membrane damages, and nutritional imbalance in plants [76] and ultimately affect plant growth and productivity [77]. Nevertheless, to adjust stress, stress-induced plant evolved mechanisms to enhance the concentration of the majority of polyphenols [78,79] and detoxify the ROS. Phenolic compounds have high antioxidant activity [80] that can scavenge reactive oxygen species [81], and this observation lies behind the high level of total phenols in the C 50 treatment, reflecting the eroded soil without microorganisms amendment. Soil microbes transform phenolics into compounds, which help in element mineralization [9]. Phenolic compounds improve nutrient uptake through chelation of metallic ions, enhanced active absorption sites, and soil porosity with accelerated mobilization of many elements [82]. This observation can explain a positive

correlation between antioxidant activity, total phenolics, and some elements mentioned in the present study which were corroborative to the previous findings [83].

Another aspect of the present research is the translocation of mineral elements to shoots. POX catalyzes lignin formation and establishes structural barriers by producing reactive oxygen and nitrogen species [84]. This explains the high activity of GPOX in roots of tomato seedlings, confirmed for all treatments, except for C 50, where stress conditions could interrupt this line of defense. Species of *Rhizobacteria*, *Azospirillum*, and *Pseudomonas* play a significant role in tomato competition for nutrients or space [85]. Thus, inoculation of tomato with *A. brasilense* could cause increased GPOX activity in inoculated treatments with the substrate with a low amount of organic matter (AMF + AZ 50 and AMF + AZ 70). Accordingly, Islam et al. [86] stated that PGPB could activate plant antioxidant defense by regulating the activity of superoxide dismutase, catalase, and peroxidase, the key enzymes that deactivate over produced reactive oxygen species.

## Element concentration in plant tissues

We analyzed the substrate content of available forms of nutrients and the uptake by tomato roots followed by translocation to shoots. Plants need to develop efficient strategies to enhance K uptake from the soil, which includes association with soil microbiota [87].

According to Meena et al. [88] the most effective K, P, and Zn-solubilizing bacteria belong to the genera *Azospirillum*, *Azotobacter*, and *Bacillus*. The results concerning potassium substrate/root/shoot translocation showed an effective K collection from the peat:sand substrate, accelerated by consortia AMF + AZ 50 and AMF + AZ 70. According to Singh et al. [89], K acquisition from soils with low soluble K concentration can be enhanced by mycorrhizal symbiosis. In the present research, the negative correlation observed for K in the substrate and in tomato leaves indicated the existence of microbial-assisted release and K uptake.

Positive effects of K on dry biomass are observed; however, increasing K application can also decrease the economic benefit because excess K can reduce the mobility of calcium [90]. The negative antagonism was confirmed for K and Ca in the aboveground tomato tissues in the presented research. Free Ca level in a plant tissue is a stress signaling factor, and $Ca^{2+}$ plays a vital role in many functions. AMF selectively uptake $K^+$ and $Ca^{2+}$, which act as osmotic equivalents as they avoid the uptake of toxic $Na^+$, especially in the saline soils [91]. However, in the present research, inoculation did not have a significant effect on $Ca^{2+}$ absorption, which was dependent mainly on substrate composition. Root tissues accumulate a higher level of $Na^+$ than shoots; moreover, in mycorrhizal roots $Na^+$ may be compartmentalized in cell vacuoles and in AMF hyphae to prevent translocation to the shoots [92]. Indeed, tomato seedlings accumulated Na in roots in the conditions of the present experiment.

PGPM inoculation improved P, Mg, and Ca contents in plants. Several studies have shown that AMF helps in the P nutrition of plants to the extent of saving NPK fertilizer application with no adverse effect on growth and yield of tomatoes [93]. AMF and phosphate solubilizing bacteria (PSB) could interact synergistically because PSB solubilizes sparingly available P into orthophosphate such that AMF can absorb and transport it to the host plant [94]. Bacteria help to mineralize organic P in the soil by the synthesis of enzymes (phytases, phosphonoacetate hydrolases) [95]. Although there is some evidence concerning the effect of PGPM on P and K plant nutrition, the knowledge regarding the other elements is limited. PGPM also improved Fe concentration in tomato across all conditions, concerning low and high P availability and Mg under low P availability to plants [96]. Mg is involved in a wide range of physiological activities, including pigment synthesis, energy metabolism, and photosynthetic carbon fixation [97]. An interesting result was the higher Mg accumulation in roots and leaves of

tomato in substrate composed of peat:sand 50:50 and 50:70 (v:v) inoculated with AMF + AZ, as compared to the other inoculants applied. AMF + S inoculation significantly increased Fe and S accumulation in tomato leaves. According to available references, total accumulation of Zn, Cu, and Fe was higher, but Na was lower in mycorrhizal tomatoes grown in low-P soil (Al-Karaki, 2000). Azcón et al. [98] determined in lettuce, that with high substrate availability of P, the other nutrients level decreased. However, with a low level of P, macro- and micronutrients increased. This observation can explain negative correlations between substrates P and Ca, Mg, and Mn in roots and leaves, and Fe in roots of tomato in the present research.

## Conclusion

The improved growth and nutrient acquisition in tomatoes demonstrated the potential of AMF and bacteria colonization for protecting plants cultivated in degraded soil conditions. The establishment of the symbiosis, observed with confocal microscopy, modified the substrate conditions and involved a continuous cellular and molecular dialogue between AMF, AMF + bacteria, and plants, which included the activation of different metabolic pathways. The most spectacular effects of microorganisms in supporting the plant fitness in the conditions reflecting degrading soil covered decreased antioxidant activity and phenolic compound level as compared to the non-mycorrhizal control. This finding corresponded to the higher tolerance to stressful conditions and enhanced uptake of some nutrients followed by DW increase. However, the consortia of plant growth promoting microorganisms acted the most effectively in substrate rich in organic matter, positively shaping the parameters characterizing tomato seedling fitness. The application pro-ecological methods to improve growth and nutritional quality of tomatoes definitely include the beneficial microorganisms dedicated to positively affect plant performance.

## Supporting information

**S1 Fig. Mycelial structures (m) of fungi are mutually surrounding root hairs (rh) of tomato showing development of „interface"for the mutual flow of water and nutrients through the fungal and root grid.** The soil with higher organic matter has showed better netting. Figure was taken in sample at AMF + AZ 100 treatment. **1_2. Development of AMF spores (s) in root hairs (rh) area was also detected after inoculation in the treatment with low organic matter content (AMF + AZ 50)**. This confirms ability of AMF to colonize plant roots and form the propagative structures for further substrate colonizing. **1_3 A, 1_3B. Set up of symbiosis between AMF and tomato plants was described on arbuscules (a) structures found in root tissues**. The treatments with peat showed higher levels of arbuscules abundance. Both figures show the symbiosis created in sand:peat substrate 50:50, also (AMF + AZ 50). Bar = 20 μm.
(TIF)

**S2 Fig. Bacterial colonies (b) on tomato root hairs (rh) were found in all treatments with inoculation by *Azospirillum brasilense*.** In sterile conditions of substrate were found only these bacterial colonies as abundant (treatment AMF + AZ 70). Bar = 20 μm.
(TIF)

**S3 Fig. Bacterial colonies (b) on tomato root hairs (rh) were found in all treatments with inoculation by *Azospirillum brasilense*.** In sterile conditions of substrate were found only these bacterial colonies as abundant (treatment: AMF + AZ 100). Bar = 20 μm.
(TIF)

## Author Contributions

**Conceptualization:** Robert Pokluda, Lucia Ragasová, Andrzej Kalisz, Agnieszka Sekara.

**Data curation:** Miloš Jurica, Monika Komorowska, Marcin Niemiec.

**Formal analysis:** Lucia Ragasová, Monika Komorowska, Marcin Niemiec, Agnieszka Sekara.

**Funding acquisition:** Robert Pokluda, Agnieszka Sekara.

**Investigation:** Robert Pokluda, Lucia Ragasová, Miloš Jurica, Agnieszka Sekara.

**Methodology:** Robert Pokluda, Marcin Niemiec, Agnieszka Sekara.

**Resources:** Robert Pokluda.

**Supervision:** Robert Pokluda, Agnieszka Sekara.

**Writing – original draft:** Robert Pokluda, Monika Komorowska, Marcin Niemiec, Agnieszka Sekara.

**Writing – review & editing:** Robert Pokluda, Monika Komorowska, Marcin Niemiec, Agnieszka Sekara.

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
