## [Decision Letter · Decision Letter 0]

27 Sep 2021

PONE-D-21-25205Effects of growth promoting microorganisms on tomato seedlings in conditions of different growing mediaPLOS ONE

Dear Dr. Pokluda,

Thank you for submitting your manuscript to PLOS ONE. After careful consideration, we feel that it has merit but does not fully meet PLOS ONE’s publication criteria as it currently stands. Therefore, we invite you to submit a revised version of the manuscript that addresses the points raised during the review process.

We look forward to receiving your revised manuscript.

Kind regards,

Umakanta Sarker

Academic Editor

PLOS ONE

Reviewers' comments:

Reviewer's Responses to Questions

**Comments to the Author**

1. Is the manuscript technically sound, and do the data support the conclusions?

Reviewer #1: Yes

Reviewer #2: Yes

2. Has the statistical analysis been performed appropriately and rigorously? 

Reviewer #1: Yes

Reviewer #2: Yes

3. Have the authors made all data underlying the findings in their manuscript fully available?

Reviewer #1: Yes

Reviewer #2: Yes

4. Is the manuscript presented in an intelligible fashion and written in standard English?

Reviewer #1: Yes

Reviewer #2: Yes

5. Review Comments to the Author

Reviewer #1: Materials and methods are well defined and seems reproducible.

Consider modifying title as "Effects of growth promoting microorganisms on tomato seedlings growing in

different media conditions".

Introduction is too long, consider reducing it by 30% or three pages in total.

Likewise, discussion is too long, consider reducing it significantly and make it crisp.

Combine picture 1-4 as a new Figure and update figure numbers.

Reviewer #2: In this paper, the authors address a question of high interest for researchers studying the role of PGPM on tomato under different substrate conditions. The paper is correctly designed and carried out. Results are also consistent and very interesting. However, the current version must be revised at several points before it can be accepted. A list of the points to be addressed when revising the paper is provided below:

L35: Consider replacing “This review (Borelli et al., 2021)…” by “The review of Borelli et al. (2021)…”

L73: Replace “They” with “AMF”

L80-81: Consider re-writing the sentence in a different way. You can state that “Merrouche et al. (2017) have previously reported that…..’’

L119: Replace “little understood” by “poorly understood”

L120: Replace tomato’s scientific name with the updated one

L137: Is there any reference available which can support that easily degradable soil will be tolerated by plants when PGPM will be applied. Consider describing in more detail that PGPM can increase the tolerance of X number of soil stresses, so tomatoes will benefit from PGPM when cultivated under easily degradable soil.

L460-461: Please re-write the sentence in a more understandable way.

L573-574: As the referred element (Phosphorus) was referred at the previous page, consider refer again as P, and not as ‘this element’.

L629: Re-locate the words of the sentence to ‘both necessary and potentially toxic for plants’

L650: was your previous researched published? Can you provide the relevant reference?

L765-766: Can you provide reference for the sentence ‘Moreover, plants …deficit conditons’?

6. PLOS authors have the option to publish the peer review history of their article (what does this mean?). If published, this will include your full peer review and any attached files.

Reviewer #1: No

Reviewer #2: **Yes: **Panagiotis Kalozoumis

---

## [Author Response · Author response to Decision Letter 0]

1 Oct 2021

Reviewer #1: Materials and methods are well defined and seems reproducible.

Consider modifying title as "Effects of growth promoting microorganisms on tomato seedlings growing in different media conditions".

Answer: Title changed according reviewer

Introduction is too long, consider reducing it by 30% or three pages in total.

Answer: Length changed to the demanded size

Likewise, discussion is too long, consider reducing it significantly and make it crisp.

Answer: Discussion significantly shortened. MS was shortened by 10 pages, in total.

Combine picture 1-4 as a new Figure and update figure numbers 

Answer: Pictures 1-4 merged, numbering updated

Reviewer #2: In this paper, the authors address a question of high interest for researchers studying the role of PGPM on tomato under different substrate conditions. The paper is correctly designed and carried out. Results are also consistent and very interesting. However, the current version must be revised at several points before it can be accepted. A list of the points to be addressed when revising the paper is provided below:

L35: Consider replacing “This review (Borelli et al., 2021)…” by “The review of Borelli et al. (2021)…”

Answer: excluded by shortening of Introduction

L73: Replace “They” with “AMF”

Answer: replaced

L80-81: Consider re-writing the sentence in a different way. You can state that “Merrouche et al. (2017) have previously reported that…..’’

Answer: L62 – changed according to reviewer

L119: Replace “little understood” by “poorly understood”

Answer: L78 – changed according to reviewer

L120: Replace tomato’s scientific name with the updated one

Answer: valid name added to the Methodology L 107

L137: Is there any reference available which can support that easily degradable soil will be tolerated by plants when PGPM will be applied. Consider describing in more detail that PGPM can increase the tolerance of X number of soil stresses, so tomatoes will benefit from PGPM when cultivated under easily degradable soil.

Answer: L86 ref. Inui Kishi et al. (2017) added

L460-461: Please re-write the sentence in a more understandable way.

Answer: L413-415, changed to clear statement

L573-574: As the referred element (Phosphorus) was referred at the previous page, consider refer again as P, and not as ‘this element’.

Answer: changed this element to P

L629: Re-locate the words of the sentence to ‘both necessary and potentially toxic for plants’

Answer: L580 changed

L650: was your previous researched published? Can you provide the relevant reference?

Answer: By shortening of Discussion was this part excluded

L765-766: Can you provide reference for the sentence ‘Moreover, plants …deficit conditons’?

Answer: By shortening of Discussion was relevant part excluded

---

## [Decision Letter · Decision Letter 1]

11 Oct 2021

PONE-D-21-25205R1Effects of growth promoting microorganisms on tomato seedlings growing in different media conditionsPLOS ONE

Dear Dr. Pokluda,

Thank you for submitting your manuscript to PLOS ONE. After careful consideration, we feel that it has merit but does not fully meet PLOS ONE’s publication criteria as it currently stands. Therefore, we invite you to submit a revised version of the manuscript that addresses the points raised during the review process.

ACADEMIC EDITOR:

The authors addressed all the comments raised by reviewers and both reviewers are accepted the MS. Now, the manuscript improved substantially. However, before its acceptance, the authors should address these issues again with minor revision.

-There are several missing spacing between numbers and °C, before and after the symbol “+” (numerous), and “±”, etc. throughout the whole MS including figure and Table values and captions.  Check carefully and addressed accordingly.

-Follow the citation style of PLOS one in the text and reference chapter.

Line 109: Change “120°C” to “120 °C”.

Line 190-191: Change “The DPPH scavenging activity against 2,2-diphenyl-1-picrylhydrazyl (DPPH radical) was determined in plant samples.” to “The antioxidant activity was determined in plant samples following DPPH radical (2,2-diphenyl-1-picrylhydrazyl) scavenging method (add a reference here. the author may cite Doi.10.1038/s41598-021-91157-8).”.

Line 254: Table 2: Change “mMol” to “mM”.

-Discussion needs to be improved with the mechanistic explanation of the results citing relevant references.

Line 651: To explain the reason for enhanced polyphenol synthesis add the sentences at the end of the sentence “-----(Sharma et al., 2019).” “Abiotic stresses create osmotic stress (doi.10.3389/fpls.2020.559876), oxidative damage (doi.10.1007/s12010-018-2784-5), and reactive oxygen species (ROS) (doi.10.1038/s41598-018-34944-0), that lead to numerous physio-molecular changes, including a decrease in photosynthetic activities (doi.10.1016/j.foodchem.2018.01.097), DNA, protein and membrane damages, and nutritional imbalance in plants (doi.10.1371/journal.pone.0206388) and ultimately affect plant growth and productivity (doi.10.1002/jsfa.9423). Nevertheless, to adjust stress, stress-induced plant evolved mechanisms to enhance the concentration of the majority of polyphenols (doi.10.1038/s41598-018-30897-6; doi.10.1186/s12870-018-1484-1) and detoxify the ROS."

Line 651: Phenolic compounds have high antioxidant activity (add a reference here. the author may cite doi.10.3389/fnut.2020.587257) that can scavenge reactive oxygen species (add a reference here. the author may cite (doi.10.1186/s12870-020-02780-y),

Line 109: Change “elements mentioned in the present study” to “elements mentioned in the present study were corroborative to the previous findings (add a reference here. the author may cite doi.10.1038/s41598-020-71714-3)”.

We look forward to receiving your revised manuscript.

Kind regards,

Umakanta Sarker

Academic Editor

PLOS ONE

Journal Requirements:

Reviewers' comments:

Reviewer's Responses to Questions

**Comments to the Author**

1. If the authors have adequately addressed your comments raised in a previous round of review and you feel that this manuscript is now acceptable for publication, you may indicate that here to bypass the “Comments to the Author” section, enter your conflict of interest statement in the “Confidential to Editor” section, and submit your "Accept" recommendation.

Reviewer #1: All comments have been addressed

Reviewer #2: All comments have been addressed

2. Is the manuscript technically sound, and do the data support the conclusions?

Reviewer #1: Yes

Reviewer #2: Yes

3. Has the statistical analysis been performed appropriately and rigorously? 

Reviewer #1: Yes

Reviewer #2: Yes

4. Have the authors made all data underlying the findings in their manuscript fully available?

Reviewer #1: Yes

Reviewer #2: Yes

5. Is the manuscript presented in an intelligible fashion and written in standard English?

Reviewer #1: Yes

Reviewer #2: Yes

6. Review Comments to the Author

Reviewer #1: Well it was already a worth to publish manuscript, however, I found some minor changes and authors have now carefully modified the paper. It can be published in its current form.

Reviewer #2: Authors have clearly revised the manuscript based on my comments, so this work can be accepted for publication

7. PLOS authors have the option to publish the peer review history of their article (what does this mean?). If published, this will include your full peer review and any attached files.

Reviewer #1: No

Reviewer #2: **Yes: **Panagiotis Kalozoumis

---

## [Author Response · Author response to Decision Letter 1]

17 Oct 2021

Response to reviewers October 17th 2021

According to the demands, there were done all changes, as described below.

ACADEMIC EDITOR:

The authors addressed all the comments raised by reviewers and both reviewers are accepted the MS. Now, the manuscript improved substantially. However, before its acceptance, the authors should address these issues again with minor revision.

-There are several missing spacing between numbers and °C, before and after the symbol “+” (numerous), and “±”, etc. throughout the whole MS including figure and Table values and captions. Check carefully and addressed accordingly.

ANSWER: corrected spacing by all symbols, units

-Follow the citation style of PLOS one in the text and reference chapter.

References are listed at the end of the manuscript and numbered in the order that they appear in the text. In the text, cite the reference number in square brackets (e.g., “We used the techniques developed by our colleagues [19] to analyze the data”). PLOS uses the numbered citation (citation-sequence) method and first six authors, et al.

ANSWER: corrected to rule 6 authors max, citation style corrected

Line 109: Change “120°C” to “120 °C”.

ANSWER: corrected

Line 190-191: Change “The DPPH scavenging activity against 2,2-diphenyl-1-picrylhydrazyl (DPPH radical) was determined in plant samples.” to “The antioxidant activity was determined in plant samples following DPPH radical (2,2-diphenyl-1-picrylhydrazyl) scavenging method (add a reference here. the author may cite Doi.10.1038/s41598-021-91157-8).”.

ANSWER: corrected

Line 254: Table 2: Change “mMol” to “mM”.

ANSWER: corrected

-Discussion needs to be improved with the mechanistic explanation of the results citing relevant references.

Line 657: To explain the reason for enhanced polyphenol synthesis add the sentences at the end of the sentence “-----(Sharma et al., 2019).” “Abiotic stresses create osmotic stress (doi.10.3389/fpls.2020.559876), oxidative damage (doi.10.1007/s12010-018-2784-5), and reactive oxygen species (ROS) (doi.10.1038/s41598-018-34944-0), that lead to numerous physio-molecular changes, including a decrease in photosynthetic activities (doi.10.1016/j.foodchem.2018.01.097), DNA, protein and membrane damages, and nutritional imbalance in plants (doi.10.1371/journal.pone.0206388) and ultimately affect plant growth and productivity (doi.10.1002/jsfa.9423). Nevertheless, to adjust stress, stress-induced plant evolved mechanisms to enhance the concentration of the majority of polyphenols (doi.10.1038/s41598-018-30897-6; doi.10.1186/s12870-018-1484-1) and detoxify the ROS."

Line 664: Phenolic compounds have high antioxidant activity (add a reference here. the author may cite doi.10.3389/fnut.2020.587257) that can scavenge reactive oxygen species (add a reference here. the author may cite (doi.10.1186/s12870-020-02780-y),

Line 671: Change “elements mentioned in the present study” to “elements mentioned in the present study were corroborative to the previous findings (add a reference here. the author may cite doi.10.1038/s41598-020-71714-3)”.

ANSWER: corrected

Yours faithfully

Robert Pokluda

---

## [Editor Report · Decision Letter 2]

19 Oct 2021

Effects of growth promoting microorganisms on tomato seedlings growing in different media conditions

PONE-D-21-25205R2

Dear Dr. Pokluda,

We’re pleased to inform you that your manuscript has been judged scientifically suitable for publication and will be formally accepted for publication once it meets all outstanding technical requirements.

Please addressed few typos error during proof reading:

-The authors newly include a space between number and %. Delete the space throughout the whole MS. 

-Table 1, 2, 3, 4, 5 and 6: Reduce font size to accommodate values and writing nicely.

-Line 615: Change “(e.g. [66])” to “[66]”. Also in line 616 

-Reference chapter: Number the last reference. Use this number in text also.

Kind regards,

Umakanta Sarker

Academic Editor

PLOS ONE
---

## [Editor Report · Acceptance letter]

25 Oct 2021

PONE-D-21-25205R2 

Effects of growth promoting microorganisms on tomato seedlings growing in different media conditions 

Dear Dr. Pokluda:

I'm pleased to inform you that your manuscript has been deemed suitable for publication in PLOS ONE. Congratulations! Your manuscript is now with our production department. 

Kind regards, 

on behalf of

Professor Umakanta Sarker 

Academic Editor

PLOS ONE